# Factors Associated with Utilization of Primary and Specialist Healthcare Services by Elderly Cardiovascular Patients in the Republic of Serbia: A Cross-Sectional Study from the National Health Survey 2013

**DOI:** 10.3390/ijerph17072602

**Published:** 2020-04-10

**Authors:** Andrija Grustam, Aleksandra Jovic Vranes, Ivan Soldatovic, Predrag Stojicic, Zorana Jovanovic Andersen

**Affiliations:** 1Contract Research Organization for Medical Devices & Services, 1204 Geneva, Switzerland; 2School of Public Health and Health Management, Faculty of Medicine, University of Belgrade, 11000 Belgrade, Serbia; aleksandra.jovic-vranes@med.bg.ac.rs; 3Institute of Social Medicine, Faculty of Medicine, University of Belgrade, 11000 Belgrade, Serbia; 4Institute of Medical Statistics and Informatics, Faculty of Medicine, University of Belgrade, 11000 Belgrade, Serbia; ivan.soldatovic@med.bg.ac.rs; 5The Rippel Foundation, Morristown, NJ 07960, USA; predrag.stojicic@gmail.com; 6ReThink Health, Cambridge, MA 02139, USA; 7Department of Public Health, Faculty of Health and Medical Sciences, University of Copenhagen, 1014 Copenhagen, Denmark; zorana.andersen@sund.ku.dk; 8Nykøbing F Hospital, Centre for Epidemiological Research, 4800 Nykøbing F, Denmark

**Keywords:** cardiovascular diseases, healthcare utilization, logistic regression, Serbia

## Abstract

The European Health Interview Survey (EHIS) is run every 5 years to examine how people experience and rank their health, how they care about their health, and to what extent they use the healthcare services. We identified the sub-population of special interest, i.e., cardiovascular disease (CVD) patients older than 65 years, in this cross-sectional study from the Serbian national survey of population health (2568 persons from a total of 15,999 subjects surveyed). We performed univariable and multivariable logistic regression analysis to assess the correlation between the healthcare system utilization and identified demographic, geographic, socio-economic, and self-rated factors. The most important factor for the utilization of the primary and the specialist healthcare services by elderly CVD patients is the region where one lives (Southern and Eastern Serbia OR = 2.44, 95% CI = 1.58–3.77/Belgrade OR = 1.75, 95% CI = 1.32–2.30). Age is another factor, where the 65 to 74 years old CVD patients utilize healthcare services the most. Higher education (OR = 1.80, 95% CI = 1.31–2.47), being a part of the highest Wealth Index group (OR = 1.62, 95% CI = 1.10–2.40), having very poor health status (OR = 3.02, 95% CI = 1.41–6.47), and presence of long-term illness (OR = 1.49, 95% CI = 1.16–1.92), play an important role in the utilization of the specialist care only.

## 1. Introduction

Cardiovascular diseases (CVDs) are the number one cause of death in the world. The World Health Organization estimated that in 2016 almost 17.9 million people died from CVD, corresponding to 31% of all deaths globally [1]. Of these, 7.4 million people died from coronary heart disease and 6.7 million from a stroke. Other major diseases in this group include ischaemic and rheumatic heart disease, congenital heart disease, deep vein thrombosis, and pulmonary embolism.

CVDs are chronic diseases with heavy utilization of healthcare services [2]. WHO reports on striking inequalities in access to care, utilization of care, and health outcomes in CVDs as well as other chronic diseases [3]. Three-quarters of the world deaths resulting from CVDs are in low- and middle-income countries [1]. The WHO believes that people living in those countries have reduced access to primary and specialty healthcare services, and thus do not benefit from the programs aimed at early disease detection.

The problem of inequitable utilization of healthcare is universal and pervasive. In Bulgaria, which together with Serbia transitioned from the Soviet-style model of health coverage, the population uses obscure strategies and informal payments to access advanced care services [4]. In Croatia, which was once together with Serbia a part of Yugoslavia, there are problems with access to and utilization of the healthcare system in relation to socioeconomic status [5]. Various authors have reported similar problems in the developed countries: in Spain, the utilization of curative healthcare services is found to be related to social class [6], while in Britain the use of General Practitioner (GP) is largely equitable but the specialist care is found to be “pro-rich” [7]. In Denmark, gender and gender-specific factors, socio-demographic and lifestyle factors are found to be independent determinants of frequent attendance at GP [8].

A widely used tool for studying healthcare services utilization is Andersen’s behavioral model [9]. According to this model, healthcare utilization is a function of the three factors: predisposing, enabling, and need factors. Predisposing factors (social and demographic) reflect a person’s ability to use healthcare services, enabling factors (economic) are the resources that facilitate the access, and the need factors (health outcomes) are individual needs for the healthcare service [10]. Based on this model, “equity in health service is achieved when the need factors have a strong positive association with health service utilization” [10,11] while on the other hand “enabling resources (e.g., health insurance or income) may lead to inequity in health service” [10,12].

A study performed in 2003 found that ischemic heart disease, cerebrovascular diseases, lung cancer, unipolar depressive disorders, and diabetes mellitus were responsible for more than two-thirds (70%) of the total burden of disease and injury in the Republic of Serbia [13]. The same study found ischemic heart disease to contribute the most disability-adjusted life years (DALYs) per 1000 (26.1) followed by stroke (17.1) in the male population in Serbia. They estimated a total loss of 150,886 DALYs for ischaemic heart disease and 136,090 DALYs for cerebrovascular disease every year in Serbia [13]. Another study provides a similar estimate for the burden of ischemic heart disease in Serbia in 2000: 96,023 DALYs for men and 54,866 DALY for women [14]. The government created the National Program for Prevention and Control of CVDs in the Republic of Serbia by 2020 [15] in response to these alarming figures. In 2011, 55,514 persons (25,454 males and 30,060 females) died from CVDs in Serbia [16]. Interestingly, women died more frequently due to CVDs (59%) than men (49%), when considering all mortality causes. Serbia is according to these figures a country where the risk of dying of CVDs is the highest in Europe.

The healthcare system in the Republic of Serbia is organized on two levels (primary care and specialty care) and in three tiers: primary—consisting of 158 community health centers (“dom zdravlja”), 35 state pharmacies, and 16 health departments (“zavod”); secondary—consisting of 40 general hospitals (“opsta bolnica”) and 37 specialized hospitals (“specijalna bolnica”); and tertiary—consisting of four clinical centers, four clinics, and 16 institutes. Primary physician is a “gatekeeper” of the healthcare system and refers the patient to the secondary (hospitals) and tertiary level (clinics) to be seen by a specialist. In contrast to hospitals, clinics are health institutions that perform highly specialized consultative and stationary healthcare activities within a certain branch of medicine or dentistry. Clinics also perform educational and scientific research activities in accordance with the Law on Health Care [17]. Clinics in Serbia are only to be found in the university cities with a faculty of the health profession, i.e., in Belgrade (Capital district), Novi Sad (Vojvodina, North Serbia), Kragujevac (Sumadija, Central Serbia), and Nis (Southern Serbia). The healthcare system is managed by the three instances: Ministry of Health—in charge of health policy, care standards, quality of services and control; Institute of Public Health—in charge of health-related data and statistics, national health IT systems coordination, and recommendations for improvement of population health and the healthcare system; and the National Health Insurance Fund—responsible for healthcare system financing, reimbursement negotiations with public and private providers, and the basic healthcare services coverage in the Republic. Private health insurance, which allows access to private ambulatory and hospital care, is intended for persons who have the status of an insured person under the Compulsory Health Insurance Plan of the Republic of Serbia but want additional insurance/services, or are not covered by the Plan (e.g., foreign nationals). Private health insurance is voluntary.

Total expenditure on healthcare in Serbia, as a percent of GDP, increased from 7.4% in 2000 to 10.3% in 2011 [3]. The financing of outpatient and hospital healthcare increased from 1.84% of GDP in 2004 to 2.18% of GDP in 2008 [18]. However, in the same period, there was a reduction in the financing of the preventive healthcare, rehabilitation, diagnostics, laboratory, as well as medicines and other medical devices deployed outside hospitals [18]. Since the 1990s (the breakup of Yugoslavia) private practice has been equated with the State healthcare service. However, private practice has not yet been integrated into the healthcare system, nor in the process of compulsory health and statistical reporting. In 2013, private practice was used by 15.1% of the population (out-of-the pocket payments) with no reimbursement of the expenses by the State [19]. Private practice was mostly used by residents with the highest education (27.9%), highest income (24.8%), residents of Belgrade and urban settlements [19].

Serbian population is aging fast. The median age grew from 40.2 years in 2002 to 41.2 years in 2009 [20]. The demographic projections for the period 2002–2032 indicate a depopulation [20]. According to five projections, the number of people over 65 in 2030 will account for 21% of the population, from 17.1% in 2009, while the share of the 80 years and older will increase from 3.3% in 2009 to 5.0% in 2030 [21]. The relative contribution to the elderly group, i.e., over 65 years, over the next two decades is disproportionate—while the total number of citizens is projected to drop from 7,320,807 in 2002 to 6,888,888 in 2030, the number of 65 years and older is going to increase from 1,250,818 in 2009 to 1,450,349 in 2030 [22], while the number of 80 years and older will increase from 244.579 to 344.796 persons. This is due to lower natality, and the emigration of younger people.

The burden of CVDs in Serbia is a burning issue, representing approximately 1.8% of the Serbian gross domestic product (GDP) in 2009 [23]. Apparently, the increase in healthcare expenditure did not tackle the rising costs of CVDs. These costs are due to the increase in the elderly population, but also partially a result of increased healthcare services utilization. The State’s budget is under increased pressure, while at the same time out-of-the pocket health-related expenditure contributes considerably to household impoverishment [24]. We know how much of the healthy life is lost in Serbia (expressed in DALYs) but we are not sure if the allocated portion of the budget for healthcare is enough to service the increasing elderly CVD population. One of the proxies to building this knowledge is in understanding the interaction of this special-interest group with the healthcare system. Thus, the aim of this research is to understand which factors are related with the utilization of primary and specialist healthcare services, and if there is a difference in-between the primary and specialist services utilization, in elderly (65 years and older) CVD patients in the Republic of Serbia.

## 2. Materials and Methods

This research was designed as a cross-sectional study from the latest survey of population health conducted in the Republic of Serbia in 2013. We identified the sub-population of interest, i.e., CVD patients older than 65 years, and independent variables from the survey’s database. We performed univariable and multivariable logistic regression, where the dependent variable was the healthcare system utilization, and presented the results as odds ratios (OR) with 95% confidence intervals (CIs). The study was approved by the Ethics review committee of the Faculty of Medicine, University of Belgrade, after the approval of the director of the Institute of Social Medicine, University of Belgrade.

### 2.1. National Health Survey

The Ministry of Health of the Republic of Serbia conducted the National Health Survey in the period from 7th October to 30th December 2013 [19]. The survey was carried out by the Institute of Public Health of Serbia ‘Dr. Milan Jovanovic Batut’, within the framework project Delivery of Improved Local Services (DILS) financed by the World Bank [25]. The survey included subjects 7 years or older from 6500 randomly chosen households from the entire Serbia (excluding Kosovo and Metohija). The interview with study subjects as well as the measurement of height, weight and blood pressure, was carried out by professional teams of interviewers with the support of medical workers. Participation in the survey was voluntary.

The survey questionnaire was prepared based on the recommendations by the WHO and the European Health Interview Survey carried out in the countries of the European Union [26]. The two previous surveys conducted in Serbia in 2000 and 2006 fully followed the EHIS methodology. In the third survey in 2013, the harmonization of the research instruments was performed (methodology, questionnaires, instructions) with the instruments of the EHIS wave 2 [27]. There were six major areas of concern: (1) household characteristics: number of people, size and composition of the living space, electricity, heating, running water, sanitary facilities, (2) socio-demographic characteristics: income and expenses, wealth, private possessions, leisure, (3) health: chronic non-communicable diseases, accidents and injuries, physical and sensory functional limitations, ability to perform daily activities, pain, mental health, (4) utilization of healthcare services: outpatient and hospital services, preventive checks, medication use, unfulfilled health needs, satisfaction with service, (5) health determinants: diet, physical activity, provision of informal care or help, hygienic habits, smoking, alcohol consumption, use of psychoactive substances, sexual behavior, violence, social support, and (6) basic anthropogenic measurements: height, weight, waist circumference, and blood pressure.

Survey complied with the ethical standard of the World Medical Association Declaration of Helsinki [28]. In order to respect the privacy of the research subjects, necessary steps have been taken in accordance with the Law on Protection of Personal Data [29], the Law on Official Statistics [30], and the Directive on Personal Data Protection [31]. Confidentiality of information received was ascertained by data anonymization and aggregation. Written informed consent from each participant was obtained before conducting the survey.

The main goal of the survey was to describe the health status of the population on the level of the Republic and in the four statistical regions (Vojvodina; Belgrade; Sumadija and Western Serbia; Southern and Eastern Serbia). The main purpose of this survey and EHIS is to examine how people experience and rank their health, how they care about their health, and to what extent they use the healthcare services. Data from these surveys portray changes in behavior and health habits of citizens, as well as needs associated with health and healthcare use, and are intended to be used in health policy creation [19]. The survey concerned the utilization of the public healthcare system (private practice was excluded) in the preceding twelve months.

### 2.2. Study Population

The study population was sampled from the most complete population register in Serbia—Census of Population, Households and Dwellings in the Republic of Serbia 2011, which includes a list of all households in all census enumeration circles [32]. In accordance with the EHIS wave 2, the nationally representative probability sample was used—a stratified two-tier sample with a known probability selection of sample units at each sampling stage [33]. The sample was selected to provide a statistically reliable estimate of a large number of indicators of population health, at the national, regional, and local levels. Mechanisms that were used to produce a random sample of households and respondents represent a combination of two sampling techniques—stratification and multi-stage sampling.

Out of 19,079 persons from 10,089 invited households, a total of 15,999 subjects, from 6500 households participated in the survey. In this project, we were interested in the elderly population with CVDs and identified 2568 persons who were 65 years of age and older (i.e., the retirement age for men in Serbia; 60 years is for women), and who reported having heart failure, hypertension, myocardial infarction or stroke.

### 2.3. Data and Variables

The data came in a form of three databases that we received from the Institute of Public Health of Serbia, all of which contained self-reported data. In the database containing information on citizens older than 15 years, we first distinguished between necessary and unnecessary data. We identified independent variables of interest to our research and (re)grouped them into four categories: demographic and socio-economic factors (predisposing factors), geographic factors (enabling factors), and self-rated health factors (need factors). The data used are available at Harvard Dataverse [34].

We run several models to determine which factors have an impact on the dependent variable, i.e., healthcare utilization, and subsequently included them in the univariable and multivariable analysis. We investigated how each influenced the utilization of both the primary and the specialist healthcare services. Proxy to service utilization was the engagement of healthcare staff. Primary care physicians in the survey were general practitioners and occupational therapists. Cardiologists, rheumatologists, ophthalmologists, otologists, gynecologists, neuropsychiatrists, physiatrists, internists and surgeons, irrespective of their affiliation to the secondary or tertiary level of care, were considered as specialists.

We included the following demographic factors in the final statistical model: age (65–74 years, 75–84 years, 85+ years), sex (male, female), cohabitation (yes, no), and marital status (never married, married/partnership, widow/widower, divorced). Regarding the socio-economic factors, we included in the analysis educational level (elementary school, high school, higher education), Wealth Index (grade 1 to 5), and income per member of the household, in thousands of 2014 Serbian dinars (< 9, 9–14, 15–19, 20–29, > 29, refused to disclose). The Demographic and Health Survey Wealth Index [35] was constructed from the assets under one’s control, e.g., number of bedrooms in the household, materials used, access to drinking water and sanitation, source of energy, possession of household appliances, computers, cell phones, access to the Internet. The principal component analysis was used to assign weights to each variable. The indicator values were multiplied by the loadings and summed to produce the household value, i.e., the Wealth Index [33]. Distribution of respondents from the lowest to the highest values of the household index to five categories (quintiles) was carried out, so in the first quintile the poorest citizens were grouped and in the fifth the richest.

Regarding the geographic factors, we included the following variables in the model presented in this paper: the region (Northern Serbia, Capital district, Central and Western Serbia, Southern and Eastern Serbia), and the type of settlement (city/urban, non-urban). We were also interested in knowing if there is a long distance from the healthcare services (yes, no, have not utilized health care services in the last 12 months), and a long wait for the provision of the service (yes, no, have not utilized health care services in the last 12 months) both self-defined and self-reported. Furthermore, we were interested in the personal health status of the respondents (very good, good, average, poor, very poor), the existence of a long-term illness other than a CVD (yes, no), and impairments of daily activities (serious impaired, impaired, not impaired). We also included factors on dental health (very good, good, average, poor, very poor).

### 2.4. Statistical Analysis

The results are presented as counts (percent). Pearson chi-square and Mantel–Haenszel chi-square test for trend were used to assess differences between groups. Exact tests were used, where appropriate (expected count less than 5 in more than 20% cells in crosstabulation). Logistic regression analysis was used to assess the association between healthcare system utilization and independent variables. For the multivariable model, we used only variables with a *p*-value of less than 0.1 in the univariable analysis. We have tested for the presence of multicollinearity in the multivariable logistic regression analysis using an analysis of correlations between independent variables or by a variance inflation factor. The independent variables in the model that have high inter-correlation were excluded from the analysis due to non-significant improvement of the model. The final model is a set of independent variables with no multicollinearity, which is useful in showing the difference between the utilization of primary and specialist healthcare. All *p*-values of less than 0.05 were considered significant. Model fit was assessed using Nagelkerke R Square and by Area Under the Curve (c statistics). The interaction of independent variables was not tested because no a priori interaction is assumed to influence the dependent variable. Data were analyzed using IBM SPSS Statistics for Windows, Version 20.0 (IBM Corp., Armonk, NY, USA).

## 3. Results

The majority of the studied population were women, and fit with the following description: a 65–74 old person, living in Central and Western Serbia or Southern and Eastern Serbia, in an urban settlement, belonging to the poorest or second poorest stratum, with only elementary education, married with two children on average, and living with a partner.

Regarding the demographic factors, in the univariable analysis there was no difference in the use of primary and specialist healthcare services in the past twelve months by gender, but there was in relation to age—patients older than 85 years were using the services the least (OR = 0.40, 95% CI = 0.24–0.67/OR = 0.47, 95% CI = 0.33–0.68) (Table 1 and Table 2). There was a significant association between the use of specialist healthcare (but not primary) and the marital status, with widows/widowers using health services significantly less than the married ones (OR = 0.78, 95% CI = 0.66-0.92).

The socio-economic factors for the utilization of specialist healthcare services in the univariable analysis are education and income, where the highly educated subjects (OR = 1.86, 95% CI = 1.42–2.43) and those with the highest income per household member (OR = 1.42, 95% CI = 1.05–1.91) and in the best financial situation (OR = 1.98, 95% CI = 1.44–2.72) were more likely to use the specialist care (Table 2). There was no significant association between education and income with the utilization of the primary healthcare services.

The most important geographic determinant of the utilization of both the primary and the specialist care in Serbia, in the univariable analysis, was the region of residence (Table 1 and Table 2). People living in Southern and Eastern Serbia use primary care the most (OR = 2.62, 95% CI = 1.71–4.01), while people living in Belgrade use specialist care the most (OR = 2.12, 95% CI = 1.65–2.73). The type of a settlement (urban vs non-urban) was not a significant factor in primary healthcare utilization, in contrast to specialist care, with significantly lower utilization in non-urban areas (OR = 0.71, 95% CI = 0.61–0.84). Furthermore, distance did not influence the utilization of either primary or specialist care, while the long wait was found to be connected to the specialist care only.

Self-rated health factors of utilization show a similar pattern for the utilization of primary and specialist services, in the univariable analysis (Table 1 and Table 2). Self-rated poor health was associated with the utilization of both primary and specialist care (OR = 2.95, 95% CI = 1.10–7.96 / OR = 2.57, 95% CI = 1.29–5.11), while impairment of daily activities was associated with increased utilization of primary care (OR = 1.84, 95% CI = 1.30–2.60) and serious impairment with the utilization of specialist care (OR = 2.71, 95% CI = 2.17–3.39). The presence of long-term illness was associated with the utilization of specialist care only (OR = 1.83, 95% CI = 1.45–2.30).

In the multivariable analysis (Table 3), regarding the primary healthcare, age and region were the only statistically significant factors of utilization. As in the univariable analysis, the oldest (85+) used both primary and specialty services the least (OR = 0.39, 95% CI = 0.22–0.67/OR = 0.46, 95% CI = 0.31–0.68), people living in Southern and Eastern Serbia used primary services the most (OR = 2.44, 95% CI = 1.58–3.77) while people living in the capital used the specialty services the most (OR = 1.75, 95% CI = 1.32–2.30). Other significant factors for the utilization of specialist healthcare in the Republic of Serbia are higher education (OR = 1.80, 95% CI = 1.31–2.47), excellent financial situation (OR = 1.62, 95% CI = 1.10–2.40), very poor health status (OR = 3.02, 95% CI = 1.41–6.47), and presence of long-term illness (OR = 1.49, 95% CI = 1.16–1.92).

Model fit, using Nagelkerke R square and c statistic (area under the curve), reveals better fit for specialist healthcare utilization compared to GP. Model fit parameters are Nagelkerke R square = 0.067 with c statistic 0.667 (95% CI 0.629–0.706) for GP and Nagelkerke R square = 0.107 with c statistic 0.670 (95% CI 0.649–0.692) for specialist healthcare utilization.

## 4. Discussion

We found that among cardiovascular disease patients older than 65 in the Republic of Serbia, people in the age group 65–74 years and those living in southern and eastern Serbia, utilize primary healthcare the most. This is due to the lesser density of specialty care in this region than in the rest of the country, so patients are forced to frequent the primary healthcare facilities. On the other hand, being in the age group 65–74 years, living in the capital district, having a higher education, and higher income, are all important factors of utilization of the specialty care. This corresponds with the higher concentration of the secondary and tertiary healthcare facilities in the capital, but also with the socio-economic ability of these patients to navigate the complexities of the healthcare system. In other words, in the conceptual framework based on Andersen’s health behavior model [9], we found that predisposing and enabling factors affected to some extent the primary healthcare utilization, while a host of predisposing, enabling and need factors affected the specialist care utilization in Serbia.

The WHO carried a survey-based project in the regions of Vojvodina, Central and Southern Serbia, and Belgrade on primary care (and specialist care) in 2010 [36]. It was executed in the framework of the 2008–2009 Biennial Collaborative Agreement between the WHO Regional Office for Europe and the Ministry of Health of the Republic of Serbia. They found the distribution of human resources to be uneven, i.e., “in some regions physicians are working for practice populations far above the national norm, while in others there is an oversupply of staff” [36]. The average reported size of the practice population of GPs was 1197 people, in all three surveyed regions. A similar finding to ours was that most of the surveyed patients (73%) were living within 20 minutes of travel from the GP. On average Serbian citizens had four GP contacts per year, with 22% resulting in referral to a specialist and 13.5% in hospital admissions. However, 11% of surveyed patients had abstained from a visit to their doctor for financial reasons [36]. The evolution of the healthcare system in Serbia is characterized by curative care being largely offered in the specialist care facilities which are to be found only in towns and cities, which affects access to care and patient’s seeking behavior.

The results of our study can be partially explained by the primary bias of the dataset. Naturally, the younger patients utilize healthcare more due to higher mortality at the later stage of the disease. This study sample only included patients with CVDs, but healthy enough to survive 65 years, which is likely related to socioeconomic status. Furthermore, living in the capital city, having higher education, and higher income are likely highly correlated. Our analysis did not control for this, as due to collinearity some variables are excluded from the multivariable analysis. For instance, variables ‘health status’ and ‘daily activities impairment’ are highly correlated, so the former was selected as a comprehensive indicator of general health. On the other hand, the ‘long-term illness’ variable implies a poor health status but does not necessarily affect healthcare utilization, and for this reason, it remained in the model. Variables ‘cohabitation’ and ‘marital status’ imply the same thing. Since cohabitation is a binary variable, then its interpretation is far easier for the final model presented in Table 3. The *p*-value and confidence interval show near statistical significance for cohabitation and the utilization of specialty care in the Republic.

The study by Jankovic, Simic and Marinkovic, performed on the data collected by the previous National Health Survey from 2006, showed similar results to ours [33]. The authors observed that “the utilization of non-preventive health services in Serbia was more frequent in advantaged social classes” and that there are “significant differences in almost all aspects of health services utilization among different socioeconomic groups, different levels of educational attainment and different gender” [33] (p. 393). Although we performed research on a specific sample from the general population (i.e., elderly CVD patients) this corresponds well with increased utilization of primary and specialty healthcare services by the most educated and wealthiest patients in our study. On the other hand, Jankovic, Simic and Marinkovic found inequality in the use of primary healthcare services by gender, in the general population [33], while we have not. This is probably because both elderly men and women have time and interest in visiting healthcare providers.

A study by Sipetic et al. explored the risk factor and the burden of the selected conditions in Serbia [37]. They found that “more than 40% of all deaths and of the total Years of life lost (YLLs) are attributable to cigarette smoking, overweight, physical inactivity, inadequate intake of fruit and vegetables, hypertension and high blood cholesterol” [37] (p. 445). For the CVDs the main risk factor is hypertension, similarly to other developing countries [38]. In Serbia, hypertension was responsible for 12% of all deaths and 13.3% of total YLLs, more in females than in males [37]. Following these results, we would expect to find the increased utilization of the healthcare services in our study by females, which we did not.

Scandinavian countries are believed to be the most equitable in the world. However, in Denmark Gundgaard found income-related inequality in utilization of health services, where the poorest citizens consume a bigger share than the rich, but when it comes to specialist healthcare services, they have significantly less drug consumption and dental treatments than expected [39]. Jørgensen et al. found that women had 18% higher rate of GP visits than men in Denmark (after adjustment for gravidity and post-menopausal hormone therapy), but the main determinants of utilization for both men and women were hypertension, mental illness, diabetes, angina pectoris, and unemployment [40]. In Sweden, Agerholm et al. found income differentials in the number of visits to doctors in favor of lower-income groups, when only controlling for age, but when controlling for health status higher-income groups were having 11-49% more visits than the lowest income group [41]. In Norway, Vikum, Krokstad, and Westin found pro-rich and pro-educated social inequalities in utilization of hospital outpatient services, while utilization of GPs and inpatient services was found to be equitable [42].

The scientific literature on healthcare utilization inequalities for elderly patients with CVD is sparse. Asthana et al. performed a scoping review on inequity in cardiovascular care in the English National Health Service and found that females and older persons are consistently being associated with lower than expected rates of access to and use of cardiovascular care, South Asian populations having higher access while black populations lower access to care [43]. They found the geographical variation in access/use to be striking, and that barriers to access erected by healthcare professionals explain their results, rather than patients’ failure to seek help in the first place. In China, Dou et al. performed a study on healthcare utilization in elderly people with CVDs and found that patients tended to use more outpatient care as they became older, while for inpatient care, the oldest patients aged over 80 years used it less than those 70–79 years old [44]. Household economic status had an influence on outpatient care utilization but showed no association with inpatient care utilization in Chinese elderly patients with CVDs. In the United States, the situation is complicated further with racial and ethnic issues in access and utilization of healthcare. Bhalotra et al. found a higher burden of risk factors and larger inequalities in receiving needed lifesaving cardiac procedures by race, ethnicity, and gender [45].

The reform of the Serbian healthcare system is long overdue [46]. Although expenditure on health services has increased in recent years [47] it is evident that the same set of factors influence the utilization of the healthcare system, as seen in the National Health Surveys from 2000, 2006, and 2013. The European Health Interview Surveys [26] are planned to be run periodically, every five years. The importance of our research is in “connecting the dots” between previous and future health surveys, thus allowing for the continuous fight against inequitable access to healthcare and for drawing comparisons with other European countries. The evidence presented in this paper, for the selected patient group which is accruing the most DALYs and the most costs (i.e., elderly CVD patients), should help the Ministry of Health in creating new policies and strategies to deal with the unequal utilization of primary and specialty healthcare in Serbia. Without a systemic approach, the identified factors will continue to breed inequalities in access and utilization of healthcare. We expect the same factors to feature again in the next national health survey, but this remains to be seen. Future research should compare our findings with the findings for the elderly CVD population in the following (and possibly preceding) EHIS, as well as with other countries where EHIS was run in order to help health policy creation based on the best practices.

### Limitations

Our analysis concerned two levels of healthcare utilization: the primary, and the specialty care which is organized in the secondary and tertiary tier. The tertiary tier in the Republic of Serbia is the highest level of healthcare service and is provided in clinics, institutes, and major healthcare centers. Our analysis did not discern between the secondary and the tertiary level as many healthcare facilities operate on both. In our analysis they are merged, which might be masking important insights. Furthermore, we did not distinguish between the public and private healthcare consumption. The private healthcare services were not in the scope of the national health survey and thus we had no data on those. The major limitation of this research is that it is based on self-reported data. The issues with self-reported data, regarding utilization of healthcare services, revolve around cognitive abilities, recall time frame, type of utilization, utilization frequency, questionnaire design, mode of data collection, and memory aids and probes [48].

## 5. Conclusions

This study confirms the previous findings of inequal and inequitable utilization of the healthcare services in the Republic of Serbia. Elderly patients with CVDs show a significant difference in frequenting the primary and specialty healthcare facilities, with age and region of residence as the most important factors of the utilization of both the primary and the specialty care. Higher education, wealth, and poor health status with a long-term illness play an important role in the utilization of specialty care only.

## Figures and Tables

**Table 1 ijerph-17-02602-t001:** Univariable analysis of the utilization of primary healthcare services by elderly CVD patients in the Republic of Serbia.

Factors	Categories	% (n) of Category	n (%) of Utilization	OR (95% CI)	*p*-Value
Age	65–74	53.8 (1338/2489)	1246/1338 (93.1)	1	
	75–84	41.1 (1023/2489)	932/1023 (91.1)	0.76 (0.56–1.02)	0.070
	85+	5.1 (128/2489)	108 /128 (84.4)	0.40 (0.24–0.67)	0.001
Sex	Male	38.1 (949/2489)	867/949 (91.4)	1	
	Female	61.9 (1540/2489)	1419 /1540 (92.1)	1.11 (0.83–1.49)	0.488
Cohabitation	No	46.2 (1151/2489)	1057/1151 (91.8)	1	
	Yes	53.8 (1338/2489)	1229/1338 (91.9)	1.00 (0.75–1.34)	0.985
Marital status	Married/partnership	53.8 (1338/2489)	1229/1338 (91.9)	1	
	Never married	1.0 (24/2489)	21/24 (87.5)	0.62 (0.18–2.11)	0.446
	Widow/widower	42.5 (1059/2489)	970/1059 (91.6)	0.97 (0.72–1.29)	0.820
	Divorced	2.7 (68/2489)	66/68 (97.1)	2.93 (0.71–12.11)	0.138
Education	Elementary school	56.9 (1415/2489)	1300/1415 (91.9)	1	
	High school	30.8 (766/2489)	700/766 (91.4)	0.94 (0.68–1.29)	0.693
	Higher education	12.4 (308/2489)	286/308 (92.9)	1.15 (0.72–1.85)	0.563
Financial situation (WI)	1	33.2 (826/2489)	748/826 (90.6)	1	
	2	22.7 (564/2489)	525/564 (93.1)	1.40 (0.94–2.09)	0.097
	3	18.4 (458/2489)	420/458 (91.7)	1.15 (0.77–1.73)	0.493
	4	16.5 (410/2489)	377/410 (92.0)	1.19 (0.78–1.82)	0.420
	5	9.3 (231/2489)	216/231 (93.5)	1.50 (0.85–2.66)	0.164
Income per household member	<9	14.8 (369/2489)	339/369 (91.9)	1	
	9–14	19.4 (483/2489)	453/483 (93.8)	1.34 (0.79–2.26)	0.279
	15–19	22.0 (547/2489)	505/547 (92.3)	1.06 (0.65–1.73)	0.803
	20–29	17.3 (430/2489)	388/430 (90.2)	0.82 (0.50–1.33)	0.421
	> 29	15.2 (379/2489)	349/379 (92.1)	1.03 (0.61–1.74)	0.914
	Refuse to disclose ^+^	11.3 (281/2489)	252/281 (89.7)	/	
Region	Northern	23.2 (587/2489)	507/578 (87.7)	1	
	Capital district	20.8 (517/2489)	473/517 (91.5)	1.50 (1.01–2.24)	0.043
	Central and Western	29.0 (723/2489)	669/723 (92.5)	1.73 (1.20–2.52)	0.004
	Southern and Eastern	27.0 (671/2489)	637/671 (94.9)	2.62 (1.71–4.01)	<0.001
Type of settlement	City (urban)	54.5 (1357/2489)	1256/1357 (92.6)	1	
	Non-urban	45.5 (1132/2489)	1030/1132 (91.0)	0.81 (0.61–1.08)	0.155
Long distance from the service	Yes	9.7 (241/2489)	227/241 (94.2)	1	
	No	82.8 (2062/2489)	1927/2062 (93.5)	0.88 (0.50–1.55)	0.660
	Not utilized	7.2 (179/2489)	128/179 (71.5)	/	
	Do not know	0.3 (7/2489)	/	/	
Long wait for the service	Yes	18.0 (449/2489)	422/449 (94.0)	1	
	No	75.8 (1887/2489)	1764/1887 (93.5)	0.92 (0.56–1.41)	0.695
	Not utilized	5.8 (145/2489)	94/145 (64.8)	/	
	Do not know	0.3 (8/2489)	/	/	
Health status	Very good	1.4 (35/2489)	30/35 (85.7)	1	
	Good	13.7 (342/2489)	297/342 (86.8)	1.10 (0.41–2.98)	0.851
	Average	38.4 (957/2489)	883/957 (92.3)	1.99 (0.75–5.28)	0.167
	Poor	35.4 (880/2489)	833/880 (94.7)	2.95 (1.10–7.96)	0.032
	Very poor	11.0 (275/2489)	243/275 (88.4)	1.27 (0.46–3.50)	0.650
Long-term illness	No	13.2 (328/2487)	293/328 (89.3)	1	
	Yes	86.8 (2159/2487)	1991/2159 (92.2)	1.42 (0.96–2.08)	0.076
Daily activities impairment	Not impaired	25.9 (644/2487)	575/644 (89.3)	1	
	Impaired	45.3 (1126/2487)	1057/1126 (93.9)	1.84 (1.30–2.60)	0.001
	Seriously impaired	28.8 (717/2487)	652/717 (90.9)	1.20 (0.84–1.72)	0.309
Dental health	Very good	2.3 (56/2484)	50/56 (89.3)	1	
	Good	13.6 (337/2484)	308/337 (91.4)	1.27 (0.50–3.22)	0.609
	Average	21.3 (530/2484)	489/530 (92.3)	1.43 (0.58–3.54)	0.437
	Poor	42.0 (1043/2484)	955/1043 (91.6)	1.30 (0.54–3.12)	0.554
	Very poor	20.9 (518/2484)	479/518 (92.5)	1.47 (0.59–3.65)	0.402

WI= Wealth Index; ^+^ Not included in analysis.

**Table 2 ijerph-17-02602-t002:** Univariable analysis of the utilization of specialist healthcare services by elderly CVD patients in the Republic of Serbia.

Factors	Categories	% (n) of Category	n (%) of Utilization	OR (95% CI)	*p*-Value
Age	65–74	54.1 (1371/2535)	880/1371 (64.2)	1	
	75–84	40.6 (1029/2535)	660/1029 (64.1)	0.10 (0.84–1.18)	0.981
	85+	5.3 (135/2535)	62/135 (45.9)	0.47 (0.33–0.68)	<0.001
Sex	Male	38.3 (970/2535)	613/970 (63.2)	1	
	Female	61.7 (1565/2535)	989/1565 (63.2)	1.00 (0.85–1.18)	>0.999
Cohabitation	No	46.4 (1175/2535)	705/1175 (60.0)	1	
	Yes	53.6 (1360/2535)	897/1360 (66.0)	1.29 (1.10–1.52)	0.002
Marital status	Married/partnership	53.6 (1360/2535)	897/1360 (66.0)	1	
	Never married	1.1 (28/2535)	17/28 (60.7)	0.80 (0.37–1.72)	0.563
	Widow/widower	42.3 (1072/2535)	644/1072 (60.1)	0.78 (0.66–0.92)	0.003
	Divorced	3.0 (75/2535)	44/75 (58.7)	0.73 (0.46–1.18)	0.197
Education	Elementary school	56.7 (1438/2535)	860/1438 (59.8)	1	
	High school	30.8 (781/2535)	510/781 (65.3)	1.26 (1.05–1.52)	0.011
	Higher education	12.5 (316/2535)	232/316 (73.4)	1.86 (1.42–2.43)	<0.001
Financial situation (WI)	1	33.2 (841/2535)	469/841 (55.8)	1	
	2	22.8 (578/2535)	362/578 (62.6)	1.33 (1.07–1.65)	0.010
	3	18.5 (469/2535)	317/469 (67.6)	1.65 (1.31–2.09)	<0.001
	4	16.4 (416/2535)	289/416 (69.5)	1.80 (1.41–2.32)	<0.001
	5	9.1 (231/2535)	165/231 (71.4)	1.98 (1.44–2.72)	<0.001
Income per household member	<9	15.0 (379/2535)	232/379 (61.2)	1	
	9–14	19.3 (488/2535)	290/488 (59.4)	0.93 (0.70–1.22)	0.594
	15–19	21.9 (555/2535)	344/555 (62.0)	1.03 (0.79–1.35)	0.813
	20–29	17.1 (434/2535)	281/434 (64.7)	1.16 (0.87–1.55)	0.298
	> 29	15.5 (392/2535)	271/392 (69.1)	1.42 (1.05–1.91)	0.021
	Refuse to disclose ^+^	11.3 (287/2535)	184/287 (64.1)	/	
Region	Northern	23.2 (589/2535)	325/589 (55.2)	1	
	Capital district	20.7 (524/2535)	379/524 (72.3)	2.12 (1.65–2.73)	<0.001
	Central and Western	29.1 (747/2535)	459/737 (62.3)	1.34 (1.08–1.67)	0.009
	Southern and Eastern	27.0 (685/2535)	439/685 (64.1)	1.45 (1.16–1.82)	0.001
Type of settlement	City (urban)	54.5 (1379/2535)	921/1379 (66.8)	1	
	Non-urban	45.6 (1156/2535)	681/1156 (58.9)	0.71 (0.61–0.84)	<0.001
Long distance from the service	Yes	9.8 (249/2535)	167/249 (67.1)	1	
	No	82.4 (2088/2535)	1361/2088 (65.2)	0.92 (0.69–1.21)	0.554
	Not utilized	7.5 (191/2535)	72/191 (37.7)	/	
	Do not know	0.3 (7/2535)	/	/	
Long wait for the service	Yes	18.1 (458/2535)	336/458 (73.4)	1	
	No	75.5 (1915/2535)	1219/1915 (63.7)	0.64 (0.51–0.80)	<0.001
	Not utilized	6.1 (154/2535)	44/154 (28.6)	/	
	Do not know	0.3 (8/2535)	/	/	
Health status	Very good	1.3 (34/2535)	17/34 (50.0)	1	
	Good	14.1 (357/2535)	191/357 (53.5)	1.15 (0.57–2.33)	0.696
	Average	38.4 (974/2535)	552/974 (56.7)	1.31 (0.66–2.59)	0.442
	Poor	35.0 (888/2535)	639/888 (72.0)	2.57 (1.29–5.11)	0.007
	Very poor	11.1 (282/2535)	203/282 (72.0)	2.57 (1.25–5.28)	0.010
Long-term illness	No	13.4 (340/2532)	172/340 (50.6)	1	
	Yes	86.6 (2192/2532)	1429/2192 (65.2)	1.83 (1.45–2.30)	<0.001
Daily activities impairment	Not impaired	26.1 (661/2533)	331/661 (50.1)	1	
	Impaired	45.3 (1147/2533)	739 /1147(64.4)	1.81 (1.49–2.19)	<0.001
	Seriously impaired	28.6 (725/2533)	530/725 (73.1)	2.71 (2.17–3.39)	<0.001
Dental health	Very good	2.3 (58/2530)	38/58 (65.5)	1	
	Good	13.6 (345/2530)	214/345 (62.0)	0.86 (0.48–1.54)	0.612
	Average	21.2 (536/2530)	354/536 (66.0)	1.02 (0.60–1.81)	0.936
	Poor	41.8 (1058/2530)	676/1058 (63.9)	0.93 (0.53–1.62)	0.802
	Very poor	21.1 (533/2530)	317/533 (59.5)	0.77 (0.44–1.36)	0.373

WI= Wealth Index; ^+^ Not included in analysis.

**Table 3 ijerph-17-02602-t003:** Multivariable analysis of the utilization of primary and specialist healthcare services by elderly CVD patients in the Republic of Serbia.

Factors	Categories	Primary Care		Specialist Care	
		OR (95% CI)	*p*-Value	OR (95% CI)	*p*-Value
Age	65–74	1		1	
	75–84	0.69 (0.51–0.95)	0.025	0.10 (0.83–1.19)	0.964
	85+	0.39 (0.22–0.67)	0.001	0.46 (0.31–0.68)	<0.001
Cohabitation	No	1		1	
	Yes	0.85 (0.62–1.16)	0.315	1.18 (0.99–1.41)	0.067
Education	Elementary school	1		1	
	High school	0.90 (0.63–1.30)	0.579	1.16 (0.94–1.43)	0.157
	Higher education	1.22 (0.71–2.11)	0.465	1.80 (1.31–2.47)	<0.001
Financial situation (WI)	1	1		1	
	2	1.25 (0.81–1.91)	0.314	1.27 (1.00–1.62)	0.047
	3	1.07 (0.67–1.69)	0.777	1.52 (1.16–2.00)	0.003
	4	1.13 (0.67–1.89)	0.653	1.61 (1.18–2.18)	0.002
	5	1.54 (0.78–3.04)	0.217	1.62 (1.10–2.40)	0.015
Region	Northern	1		1	
	Capital district	1.26 (0.82–1.94)	0.290	1.75 (1.32–2.30)	<0.001
	Central and Western	1.69 (1.15–2.48)	0.007	1.27 (1.01–1.61)	0.041
	Southern and Eastern	2.44 (1.58–3.77)	<0.001	1.34 (1.06–1.70)	0.015
Type of settlement	City (urban)	1		1	
	Non-urban	0.74 (0.51–1.06)	0.103	0.88 (0.72–1.09)	0.251
Health status	Very good	1		1	
	Good	0.10 (0.36–2.77)	0.997	1.13 (0.54–2.36)	0.737
	Average	1.73 (0.63–4.74)	0.283	1.33 (0.65–2.72)	0.440
	Poor	2.71 (0.96–7.63)	0.059	3.01 (1.45–6.24)	0.003
	Very poor	1.20 (0.42–3.45)	0.737	3.02 (1.41–6.47)	0.004
Long-term illness	No	1		1	
	Yes	1.14 (0.75–1.73)	0.534	1.49 (1.16–1.92)	0.002

WI = Wealth Index.

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
