# Peer review of "Factors Associated with Utilization of Primary and Specialist Healthcare Services by Elderly Cardiovascular Patients in the Republic of Serbia: A Cross-Sectional Study from the National Health Survey 2013"

_ijerph, 2020, doi:10.3390/ijerph17072602_

Round 1
Reviewer 1 Report
The cross-sectional study reported the findings from a secondary data analysis of the European Health Interview Survey that was conducted in 2013. A number of significant factors were found to be significantly associated with primary and specialist health care services utilization. The paper will benefit from a thorough revision with attention to the following suggestions.
- Title: The wording “influencing” suggests a cause and effect relationship, which cannot be established in a cross-sectional study. The author may want to consider change the title to “Factors associated with…..”
- Title: The title should indicate that the data is obtained from a national survey and the study design is cross-sectional.
- Abstract: The word “predictor” should be avoided in the abstract and the main text because of the nature of the cross-sectional study design.
- Abstract: The number of survey respondents should be provided.
- Methods (Line 191): Please revise the cut-points so that the categories are not overlapping (14 & 20).
- Methods (Line 208): Why did the results for risk health behavior not include in Table 1?
- Methods (Line 215-6): “Variables” should be used instead of “predictors”.
- Methods (Line 216): The authors should indicated that they have tested for the presence of multicollinearity in the multivariable logistic regression analysis.
- Methods (Line 217): SPSS 20.0” should be described as “IBM SPSS Statistics for Windows, Version 20.0 (IBM Corp., Armonk, NY).”
- Results (Line 225-227): The description is incorrect – divorced respondents did not use significantly health services more than the married respondents.
- Discussion (Line 290): The abbreviation for Years of life lost (YLLs) should be defined at its first use.
- Table 1: The layout needs to be improved. The text in the leftmost column should be consistently aligned – it is both top-aligned for age and sex etc., but middle-aligned for financial situation and income, making it difficult to read.
- Table 1 and 2: Odds ratios and 95% CIs should be reported rounded to two decimal places.
- Table 1 and 2: Exact p values should be given rather than just indicating whether it is < 0.05.
- Table 1: The denominator for each of the categories should be given. For example, what is the number of respondents in 65-74, 75-84, and ≥ 85 years should be provided.
- The percentages should be used to indicate the distribution between the categories. For example, the percentage associated with 867 males should be 37.9% (867/2286).
- Table 2: Some of the asterisks for p-value were incorrectly assigned. For example, in high school education for specialist care, with a 95% CI that includes 1 (0.943-1.436), the p value cannot be less than 0.05.
- References (Line 374): Hyperlinks in all references should be rechecked to ensure they are currently active, and if yes, the accessed date should be updated accordingly.
- References (Line 383): Sentence case instead of title case should be used for the article’s title.
- References (Line 467): Reference 42 is not a scholarly reference source.
Author Response
Dear Reviewer,
We want to thank you for your time and effort in improving our manuscript.
Please find a point-by-point response to your comments in the attachment. We hope that you will find them satisfactory.
Yours faithfully,
Andrija Grustam
Aleksandra Jovic Vranes
Ivan Soldatovic
Predrag Stojicic
Zorana Jovanovic Andersen

Reviewer 2 Report
The authors clearly state the rationale for the study and the guidance in following the EHIS methodology. The description of the study population is adequate, and I do not believe that additional detail is needed. I would like to have the authors present the significant odds ratios and their values. In the scheme of things some variables are more important than others. This would provide more context for the Discussion as well.
My preference for descriptive statistics when there are a considerable number of variables (Table 1) and variable categories is to place it in an Appendix as it is somewhat difficult to discern what important points the authors wish to make.
The issue of inequalities or disparities in healthcare based on age/condition type is clearly a public health concern and the authors make for some interesting observations with a small but select studies in the Discussion section concerning CVD. The utilization of health services by the elderly is much broader than CVD. I would be interested in the authors’ opinions in this regard as to how this study is relevant to other chronic conditions of the elderly and the implications for public health.
What the study lacks is an overall framework for viewing the findings from both a theoretical and public policy perspective. For example, the Feb. 3, 2015 IJER&PH article may be of use – Access Disparity and Health Inequality of the Elderly Unmet Needs and Delayed Healthcare. In addition, the authors might find Andersen’s behavioral model of health service utilization useful as well.
Author Response
Dear Reviewer,
We want to thank you for your time and effort in improving our manuscript.
Please find a point-by-point response to your comments in the attachment. We hope that you will find it satisfactory.
Yours faithfully,
Andrija Grustam
Aleksandra Jovic Vranes
Ivan Soldatovic
Predrag Stojicic
Zorana Jovanovic Andersen

Reviewer 3 Report
This paper looks at factors related utilization of primary and specialist healthcare services among elderly cardiovascular patients in the Republic of Serbia. The study found age, cohabitation, higher education, Wealth Index, and living area might be associated with the utilization of primary and specialist healthcare services. The findings may have significant clinical and public health implications, while there are still several issues to address:
- The authors provided comprehensive information on the burden of CVD in Serbia such as DALYs and healthcare expenditure. However, the aim of this research may not be fully supported by the information authors provided. The authors might add a small paragraph to provide a linkage to describe why they want to explore the effect of factors on primary and specialist
- The authors mentioned the healthcare system in the Republic of Serbia. International readership might not know what insurance programs are in the Republic of Serbia. Insurance programs might be associated with access to care. The authors would need to describe this briefly.
- Did the survey provide participant’s insurance types and their health behavioral variables such as smoking status and alcohol intake?
- The survey was conducted by stratification and multi-stage sampling. Did the authors consider survey weight when they run analyses?
- Please describe the multi-collinearity in “2.4. Statistical analysis.”
- Please describe the model fitness and interaction for multiple variable analyses.
- The numbers showed in the tables are hard to read. The heading of each variable is not aligned properly to the top in Tables 1 and 2.
- The statement “…the marital status, with divorced subjects using health services more than the married ones. (Line 227-228)” might be an incorrect interpretation. In table 1, divorced subjects showed the OR is 0.733 (0.456- 1.176)
- The authors might show some ORs in the paragraphs to help readers understand the effect.
- The authors mentioned predictors with a p-value of less than 0.1 in the univariable analysis were selected into multiple variable analyses. However, two logistic regression models (primary care and specialist care) were created using the same variables. The authors might need to explain this.
- Some results might need to be checked in table 2. For example, the authors showed patients aged 75-84 were significantly less likely to use specialist care (OR=0.996) compare to those aged 65-74. However, the 95% CI is from 0.831 to 1.194.
- The authors might discuss the density of physicians (primary and specialist) across the country (Northern Serbia, Capital district, Central and Western Serbia, Southern and Eastern Serbia) because it will affect the access of care and patient’s seeking behavior.
Author Response

(The authors gave the same response as above.)

Round 2
Reviewer 1 Report
The authors have satisfactorily responded to all my previous comments and made the necessary changes to the manuscript.
However, there are two minor points that should be addressed.
- Please keep the decimal place consistent (2 decimal place) for all odds ratios and their 95% CI in abstract, main text, and the tables. For example, "(Southern and Eastern Serbia OR=2.44, 95% CI=1.58-3.77 / Belgrade OR=1.75, 95% CI=1.32-2.3)" (Line 34) should be "(Southern and Eastern Serbia OR=2.44, 95% CI=1.58-3.77 / Belgrade OR=1.75, 95% CI=1.32-2.30)"
- In Table 2, please replace the p value for sex "1.000" to ">0.999.
Author Response
Dear Reviewer,
We are glad that we have satisfactorily responded to all your previous comments and made the necessary changes in the manuscript. Hereby we provide answers to your new comments.
Comment 1
In the resubmitted version of the text, we have kept the decimal place consistent (2 decimal places) for all odds ratios and their 95% CI in the abstract, main text, and the tables.
Changes made to the text, Abstract, line 34: "CI=1.32-2.30"
Changes made to the text, Abstract, line 35: "OR=1.80"
Changes made to the text, Abstract, line 36: "CI=1.10-2.40"
Changes made to the text, 3. Results, line 245: "OR=0.40"
Changes made to the text, 3. Results, line 275: "CI=1.10-7.96"
Changes made to the text, 3. Results, line 277: "CI=1.30-2.60"
Changes made to the text, 3. Results, line 279: "CI=1.45-2.30"
Changes made to the text, 3. Results, line 284: "CI=1.32-2.30"
Changes made the text, 3. Results, line 286: "OR=1.80" and "CI=1.10-2.40"
Changes made to the text, Table 3, Region, Capital District: "1.32-2.30"
Comment 2
We have replaced the p value for "sex" in Table 2. We have double checked all other p values reported in the manuscript for the same mistake.
Changes made to the text, Table 2, Sex, Female: ">0.999"
We hope that you will find the abovementioned corrections to be satisfactory.
Thank you for your time and effort in improving our manuscript.
Yours faithfully,
Andrija Grustam
Aleksandra Jovic Vranes
Ivan Soldatovic
Predrag Stojicic
Zorana Jovanovic Andersen
Reviewer 2 Report
I have reviewed the changes made by the authors. The changes are substantial and well-done.
Author Response
Dear Reviewer,
We are glad that you have found the changes to be substantial and well-done.
We thank you once again for your time and effort in improving our manuscript.
Yours faithfully,
Andrija Grustam
Aleksandra Jovic Vranes
Ivan Soldatovic
Predrag Stojicic
Zorana Jovanovic Andersen
Reviewer 3 Report
The authors addressed all the comments I provided in the last communication. I do have no further comments.
Author Response
Dear Reviewer,
We are glad that we have addressed all the comments you provided in the last communication.
We want to thank you once again for your time and effort in improving our manuscript.
Yours faithfully,
Andrija Grustam
Aleksandra Jovic Vranes
Ivan Soldatovic
Predrag Stojicic
Zorana Jovanovic Andersen